# Personalized Selection of a CFTR Modulator for a Patient with a Complex Allele [L467F;F508del]

**Elena Kondratyeva \*, Nataliya Bulatenko, Yuliya Melyanovskaya** ID **, Anna Efremova, Elena Zhekaite, Viktoriya Sherman** ID **, Anna Voronkova, Irina Asherova, Alexander Polyakov, Tagui Adyan, Valeriia Kovalskaia** ID **, Tatiana Bukharova** ID **, Dmitry Goldshtein and Sergey Kutsev**

Research Centre for Medical Genetics, Moscow 115522, Russia
* Correspondence: mucoviscidozRF@med-gen.ru; Tel.: +7-499-612-86-07

**Abstract:** The presence of complex alleles in the CFTR gene can lead to difficulties in diagnosing cystic fibrosis and cause resistance to therapy with CFTR modulators. Tezacaftor/ivacaftor therapy for 8 months in a patient with the initially established F508del/F508del genotype did not lead to an improvement in her condition—there was no change in spirometry and an increase in the patient's weight, while there was only a slight decrease in NaCl values, measured by a sweat test. The intestinal current measurements of the patient's rectal biopsy showed no positive dynamics in the rescue of CFTR function while taking tezacaftor/ivacaftor. The assumption that the patient had an additional mutation in the cis position was confirmed by sequencing the *CFTR* gene, and the complex allele [L467F;F508del] was identified. Based on the rescue of CFTR function by elexacaftor/tezacaftor/ivacaftor obtained using forskolin-induced swelling on intestinal organoids, the patient was prescribed therapy with this targeted drug. The use of elexacaftor/tezacaftor/ivacaftor for 7 months resulted in a significant improvement in the patient's clinical condition.

**Keywords:** cystic fibrosis (CF); complex allele; targeted therapy; intestinal current measurements (ICM); intestinal organoids; CFTR modulators





## 1. Introduction

Cystic fibrosis is a common hereditary autosomal recessive disease caused by pathogenic variants of the *CFTR* gene. Genetic variants of *CFTR*, according to how they decrease CFTR protein synthesis, function or stability, are usually divided into seven classes [1]. Therapy with CFTR modulators is the most effective method of treating cystic fibrosis (CF). The choice of the drug is based on the genotype and age of the patient. Currently, four targeted drugs are used to treat patients with cystic fibrosis: ivacaftor, lumacaftor/ivacaftor, tezacaftor/ivacaftor elexafactor/tezacaftor/ivacaftor (brand names Kalydeco®, Orkambi®, Symdeco® and Trikafta®, respectively) [2]. However, it has been noted that patients with an identical genotype receiving therapy with the same CFTR modulator may have a different therapeutic effect, up to a complete absence of effects. One of the reasons for this is the presence of an additional cis variant on one of the alleles of the *CFTR* gene. Meanwhile, the majority of patients with frequent mutations in the genotype, including homozygotes according to F508del, as a rule, are not examined for the carriage of other cis variants.

For the first time, complex alleles were described in an article by Kalin et al. in 1992 [3]. Each pathogenic variant as part of a complex allele can affect individual stages of CFTR protein synthesis. Complex alleles complicate the classification of *CFTR* gene mutations and can be considered disease-causing, neutral, or altering the effectiveness of treatment [4]. In patients with the F508del/F508del genotype, there is a strong variability in responses to lumacaftor/ivacaftor therapy [5,6], which may be due to the presence of cis variants.

The case discussed in this publication served as the beginning of a mass examination of Russian patients with CF homozygotes by F508del for the presence of the L467F variant

in the cis position. Previously, the complex allele [L467F;F508del] was detected in Italy, the USA, and France, where its prevalence was very low. Studies conducted in the Russian Federation with the beginning of the mass distribution of targeted therapy in the country have shown it to have a high frequency in the Russian population of homozygote F508del. It comprised more than 8.2% (data not published).

This case also demonstrates the use of modern functional methods for assessing the function of the chloride channel—the determination of intestinal current measurement (ICM) and using a forskolin-induced swelling assay on intestinal organoids to assess the effectiveness of CFTR modulators. These methods have been used in the Russian Federation since 2018 at the Research Centre for Medical Genetics to study the pathogenicity of *CFTR* gene mutations not described in international databases in complex diagnostic cases, as well as to evaluate the effectiveness of targeted therapy for carriers of rare pathogenic variants [7–9]. Forskolin-induced swelling (FIS) assay of patient-derived organoids has been used to assessment of residual functional activity of the CFTR protein and to study the effects of CFTR modulators. Forskolin activated the CFTR channel located at the apical membrane of the epithelial cells. Organoid swelling occurs through excretion chloride ions and $H_2O$ into the organoid lumen [7]. The swelling of organoids in response to forskolin is directly dependent on the functional activity of CFTR [8].

## 2. Materials and Methods

### 2.1. Molecular Genetic Analysis

The patient was a girl born in 2005 with the diagnosis of cystic fibrosis, mixed form, severe course, genotype F508del/F508del (c.[1521_1523delCTT];[1521_1523delCTT]). The diagnosis was established according to criteria in [10]. The patient was observed in the Department of Cystic Fibrosis of the Research Clinical Institute for Childhood in the Moscow region. The study and informed voluntary consent forms were approved by the Ethics Committee of the Ministry of Education and Science of the Russian Federation on 15 October 2018 (the chairman of the Ethics Committee is Professor L. F. Kurilo).

Initially, the patient's genotype was established by DNA testing for frequent mutations in the *CTFR* gene in the Russian Federation (NM_000492.4) by analysis of polymorphism of amplification fragment lengths (PDAF) and multiplex sample-dependent ligase reaction (MLPA). The patient's material was analyzed for the presence of 30 pathogenic genetic variants responsible for the development of cystic fibrosis: c.54-5940_273+10250del21kb (CFTRdele2,3), c.1521_1523delCTT (F508del), c.2051_2052delAAinsG (2183AA>G), c.1545_1546delTA (1677delTA), c.2012delT (2143delT), c.2052dupA (2184insA), c.262_263delTT (394delTT), c.3691delT (3821delT), c.413_415dupTAC (L138ins), c.1624G>T (G542X), c.3846G>A (W1282X), c.3909C>G (N1303K), c.1000C>T (R334W), c.3718-2477C>T (c.3717+12191C>T; 3849+10kbS>T), c.472_473insA (604insA), c.3816_3817delGT (3944delGT), c.3587C>G (S1196X), c.489+1G>T (621+1G>T), c.274G>A (E92K), c.3140-26A>G (3272-26A>G), c.3883delA (4015delA), c.3891dup (4022insT), c.3844T>C (W1282R), c.2657+5G>A (2789+5G>A), c.3140-16T>A (3272-16T>A), c.1397C>G (S466X), c.1766+1G>A (1898+1G>A), c.2988+1G>A (3120+1G>A), c.1040G>C (R347P), and c.2834C>T (S945L). At the first stage, the original oligonucleotide samples were annealed with the denatured DNA under study in the presence of a thermostable DNA ligase for 1 h at a temperature of 62 °C in a volume of 5 µL of the reaction mixture of the following composition: 100 fmol/µL each of oligonucleotide (Eurogen, Moscow, Russia); 0.4 units of Pfu-DNA ligase activity (Helicon, Moscow, Russia), a buffer for ligation; and 10–50 ng of genomic DNA. In the second stage, standard PCR was performed with oligo-primes complementary to sequence sites specially synthesized in oligonucleotide samples. The detection of the result was carried out using electrophoresis in polyacrylamide gel, followed by gel staining in a solution of ethidium bromide and data visualization by gel-documenting system GelDoc® 2000 in UV radiation. However, further diagnostic searches required a complete analysis of the CFTR gene. The final molecular genetic diagnosis was established by analyzing the patient's DNA on a new generation sequencer (Ion S5). Ultra-multiplex PCR technology, coupled with subsequent

sequencing (AmpliSeq), was used for sample preparation. The analysis was carried out using a custom panel, "cystic fibrosis", which includes the entire coding sequence of the CFTR gene, sections of exon-intron compounds and the region of intron 22 with mutation 3849+10kbC>T. Sequencing data processing was carried out using a standard automated algorithm offered by Thermo Fisher Scientific (Torrent Suite), as well as NGS-data (Russia, Moscow, 2021.1) software. To estimate the population frequencies of the identified variants, samples of the "1000 Genomes" and Genome Aggregation Database (gnomAD) projects were used. To assess the clinical relevance of the identified variants, the OMIM database, the HGMD® Professional version 2020.4 pathogenic variants database, the specialized CFTR2 database and literature data were used.

### 2.2. Intestinal Biopsy

Biopsy samples were taken by an endoscopist using Olympus Disposable EndoTherapy EndoJaw Biopsy forceps (model #FB-23OU) equipment, according to the instructions. The size of the biopsy was about 3–5 mm. The biopsy material was placed in Dulbecco's phosphate-saline buffer (DPBS) for ICM and culture medium with antibiotics for organelles, then transported to the laboratory.

### 2.3. Intestinal Current Measurement (ICM) Method

The ICM method was conducted according to the European standard operating procedure V2. 7_26. 10. 11 (SOPam) [11]. In the first stage, each of the four recirculation chambers was calibrated separately on the VCC MC 8B421 Physiological Instrument (San Diego, CA, USA). Physical factors were considered, such as the presence of air in the contact tips with agar and the resistance of the liquid, as well as environmental factors, such as the absence of vibrations near the equipment, accidental contacts with the electrodes and the absence of extraneous working devices in the room. In the second stage, after calibration of the device, the rectal biopsy material was placed in the chamber. The biopsy material was placed in a special slider P2407B with a diaphragm 1.2 mm in diameter, which was then inserted into the camera. All reagents were from Sigma-Aldrich, St. Louis, MO, USA. The chambers were filled with Meyler buffer solution. The buffer was prepared before the study and included the following: 105 mM NaCl, 4.7 mM KCl, 1.3 mM $CaCl_2 \cdot 6H_2O$, 20.2 mM $NaHCO_3$, 0.4 mM $NaH_2PO_4 \cdot H_2O$, 0.3 mM $Na_2HPO_4$, 1.0 mM $MgCl_2 \cdot 6H_2O$, 10 mM HEPES, and 10 mM D-glucose, as well as 0.01 mM indomethacin. The biopsies were heated to 37 °C using a circulation pump connected to a temperature-controlled water bath and continuously carbonated with 95% $O_2$ and 5% $CO_2$. The registration of the study began with the recording of the basal short-circuit current (pre-amiloride stage). At the third stage, stimulators (Sigma-Aldrich, Merck & Co., Inc., Kenilworth, NJ, USA) were added in the following sequence: amiloride (100 mM), forskolin (10 mM)/IBMX (100 mM), genistein (100 mM), carbachol (100 mM), DIDS (100 mM) and histamine (100 mM). The study was completed after the basal short-circuit current was recorded. The control group included healthy volunteers. Patients with cystic fibrosis homozygous for F508del were included in the comparison group (F508del/F508del) [12]. The study was conducted in the Scientific and Clinical Department of Cystic Fibrosis at the Research Centre for Medical Genetics (Head—Prof. Kondratyeva E.I.).

### 2.4. Obtaining Intestinal Organoids

When obtaining cultures of intestinal organoids and performing the FIS assay, protocols and articles developed and written under the guidance of J. Beekman (J.M. Beekman, Molecular Cystic Fibrosis Laboratory of the University Medical Center, Utrecht, The Netherlands) were used as a basis [13–15]. To obtain conditioned media, a mouse L-fibroblast line transfected with a Wnt-3A-expressing vector, a HEK293 cell line transfected with a Noggin-expressing vector (both lines were kindly provided by J. Beckman) and HEK293T–R-spondin-1 mFc cells line transfected with R-spondin-1-expressing vector (Cat# 3710-001-K, Trevigen) were used. All stages of cultivation were carried out at 37 °C

and 5% $CO_2$. Three cell lines were grown in DMEM+GlutaMAX medium (Thermo Fisher Scientific, Waltham, MA, USA) containing 10% embryonic calf serum (ECS; PAA Laboratories, Pasching, Austria), and penicillin/streptomycin (25,000 units and 25 mg per 500 mL medium, respectively; PanEco, Moscow, Russia). Wnt-3A conditioned medium (WCM) and Noggin conditioned medium (NCM) were produced according to protocols [15] The NEK293T cell line expressing R-spondin 1 was stably transfected and did not require cultivation on selective antibiotics. Cell culture flasks with an area of 162 cm$^2$ (Corning, corning, NY, USA) were used for cultivation. When the cells are ~80% confluent (after 3–4 days), the medium was replaced with Advanced DMEM/F12 containing 4 mM L-glutamine (PanEco, Russia) and 10 mM HEPES (PanEco, Russia). After 9–10 days of cultivation, the conditioned medium was collected in centrifuge tubes, cellular debris was precipitated ($650\times g$, 5 min) and the supernatant was transferred to new tubes. The resulting R-spondin-1 conditioned media (RCM) was stored at $-20$ °C for 1 year.

A total of 2–4 rectal biopsies are needed to isolate a sufficient number of organoids. The biopsy specimens were transported at +4 °C in a phosphate-salt buffer solution (PBS; PanEco, Russia). Isolation of crypts from biopsies was preceded by a series of washings with Advanced DMEM/F12 and PBS solution. The biopsies were then incubated with a solution of 10 mM EDTA (Thermo Fisher Scientific, USA) in PBS for 40 min at a temperature of +4 °C. Upon completion of incubation, the solution was replaced with fresh PBS and resuspended to release individual crypts from the biopsies. Then, the resulting suspension of crypts was precipitated in a centrifuge in the cooling mode for 5 min at $130\times g$ and +4 °C. The crypt sediment was mixed with "Matrigel" (Corning, USA), then seeded into 24-well culture plates (Corning, USA). For 30 min, the plates were placed in a CO2 incubator (37 °C) for polymerization of the "Matrigel" with crypts embedded in it. Then, a growth medium was added to each well. The cultivation medium contained WCM, RCM and NCM (50%, 20%, and 10%, respectively); Advanced DMEM/F12 and mEGF (50 ng/mL; Prospect); B27 (2%; Life Technologies: Gibco, Grand Island, NY USA); N-acetylcysteine (1.25 mM; Sigma-Aldrich, USA); nicotinamide (10 mM; Sigma-Aldrich, USA); A83-01 (5 μM; Tocris, Saint Louis, MO, USA); SB 202,190 (10 μM; Sigma-Aldrich, USA); and primocin (100 μg/mL; InvivoGen, Toulouse, France). Organoids needed to be reseeded about once a week. To achieve this, the growth medium was removed from all wells, the drops of "Matrigel" were mechanically destroyed and then the organoids were mechanically crushed into smaller fragments by intensive resuspending. After splitting, the resulting suspension was precipitated for 5 min at 130 g and +4 °C. The precipitate of intestinal organoids was mixed with "Matrigel", and the organoids were seeded on culture 24-well plates. The culture medium was replaced once every 2–3 days.

As a control were used two organoid cultures of CF-patient with F508del/F508del genotype and two organoid cultures of non-CF individuals [8].

### 2.5. Forskolin-Induced Swelling Assay

The forskolin-induced swelling (FIS) assay of intestinal organoids was carried out on cultures of the 3rd and higher passage. For the FIS assay. 7-days organoids were split and seeded in drops of 50% "Matrigel" into the 96-well plates (Corning, Kennebunk, ME, USA). At this stage CFTR corrector VX-661 (tezacaftor, Selleckchem, Houston, TX, USA) or VX-445 (elexacaftor, Selleckchem, USA) were added at a concentration of 3.5 μM [15] in 50 μL the cultivation medium. After 20–24 h the organoids were staining a 40-min of 0.84 μM Calcein green (Biotium, Fremont, CA, USA). The next stage added potentiator VX-770 (3.5 μM; ivacaftor, Selleckchem, Houston, TX, USA) simultaneously with forskolin (5 μM; Sigma-Aldrich, St. Louis, MO, USA) [15].

Intestinal organoids with targeted compounds in the presence of forskolin were incubated for 1 h and, simultaneously, "fixed" fields were photographed every 10 min using an Axio Observer 7 fluorescence microscope (Zeiss, Oberkochen, Germany) to register the response of organoids to stimulation. Quantitative analysis of the swelling of organoids was carried out using the Image J program and Microsoft Excel 2007 and the processing

of the obtained data was carried out using the Sigma Plot 12.5 program. The organoid swelling analysis is expressed as the absolute area under the curve (AUC) calculated from the normalized increase in surface area (baseline = 100%, t = 60 min). Replicate experiments for every condition were repeated and measurement occurred twice [15].

## 3. Results

### 3.1. Medical History

A 14-year-old female with Cystic fibrosis diagnosed at 2 years of age (positive sweat test, F508del/F508del) has pancreatic insufficiency (PI), severe pulmonary symptoms (initial lung colonization with Pseudomonas aeruginosa at the age of 6 years, bronchiectasis), polypous rhinosinusitis (poly-potomy in 2018), fatty hepatosis and chronic aspergillosis of the lungs. The patient had also recovered from a new coronavirus infection (January 2022).

Anamnesis of the disease: severe intestinal syndrome from birth and poor weight gain, up to 2 years old, were observed and treated with a diagnosis of celiac disease. After an additional examination (biopsy of the small intestine), the diagnosis of celiac disease was removed. Based on the typical clinical picture— persistent respiratory symptoms, intestinal syndrome (frequent fatty stools), lag in physical development, positive results of sweat tests (sweat conductivity—120 mmol/L NaCl), and DNA diagnostics (F508del/F508del) at the age of 2 years—a diagnosis of cystic fibrosis was established. Parents are carriers of one genetic variant in heterozygous F508del. Since 2012, there has been chronic growth of *P. aeruginosa*. In 2018, a polypotomy was performed.

Since 2017, hemoptysis has appeared with a gradual increase in volume, with to-bramycin inhalation as the provoking factor. The drug was replaced with continuous sodium colistimethate. However, in 2019, after many months of interruptions in taking sodium colistimethate, hemoptysis increased, reaching 200 mL in volume. In 2019, the bronchial arteries were embolized. Table 1 shows information on the hospitalizations of the patient and the course of the disease.

**Table 1.** Information on the hospitalization of the patient and the course of the disease for 2019–2022.

| Periods of Examination | Sweat Conductivity, NaCl mmol/L | Weight, kg | Clinical Features | FEV$_1$ |
|---|---|---|---|---|
| August 2019 | 121 | 45 | Bronchopulmonary exacerbation (increased cough, purulent sputum, hemoptysis) | 2.38 L (76.3%) |
| Start tezacaftor/ivacaftor and ivacaftor 100/150 mg and 150 mg | | | | |
| October 2019 | 114 | 46 | Examination before tezacaftor/ivacaftor and ivacaftor | 2.52 L (80.7%) |
| December 2019 | 108 | 44 | Control examination after 6 weeks of admission tezacaftor/ivacaftor and ivacaftor | 2.64 L (84.4%) |
| March 2020 | 123 | 46.5 | Bronchopulmonary exacerbation (fever, increased cough, hemoptysis) | 2.84 L (90.4%) |

**Table 1.** *Cont.*

| Periods of Examination | Sweat Conductivity, NaCl mmol/L | Weight, kg | Clinical Features | FEV$_1$ |
|---|---|---|---|---|
| March–August 2020 Break from tezacaftor/ivacaftor and ivacaftor for 1.5 months; course resumed from 25 May 2020 to 28 August 2020 | | | | |
| August 2020 | 119 | 48 | Pneumonia | 2.50 L (75%) |
| October 2020 | - | 46.5 | Respiratory viral infection | 2.91 L (87.3%) |
| September 2021 | - | 48 | Bronchopulmonary exacerbation (fever, increased cough) | 3.3 L (82%) |
| 30 November 2021 Started elexacaftor/tezacaftor/ivacaftor and ivacaftor at 100/50/75 mg and 150 mg (2 tablets in the morning, 1 tablet ivacaftor 150мг in the evening) | | | | |
| January 2022 | - | 49 | COVID-19: fever of 39 °C for 3 days, headaches | - |
| February 2022 | 55 | 54 | Pneumonia, hemoptysis CRP—21.8 mg/L | 3.84 L (108%) |
| June 2022 | 66 | 55.6 | Planned hospitalization for intravenous therapy, CRP < 5 mg/L, ESR—norm | 3.7 L (104.3%) |

For 8 months (from 24 October 2019 to 28 August 2020 with a 1.5-month break), the patient took the CFTR modulator tezacaftor/ivacaftor and ivacaftor (Symdeko®)—a pathogenetic drug for the treatment of cystic fibrosis in patients older than 12 years with the genotype F508del/F508del. Against this background, there was only a slight decrease in the indicators of the sweat test from 121 to 108 mmol of NaCl conductivity (10.7%) and the stool normalized. There were no significant positive dynamics in the parameters of spirometry; in July 2020, a pulmonary exacerbation developed, requiring hospitalization and intravenous therapy (from 30 July 2020 to 14 August 2022).

*3.2. Measure of In Vivo Response to CFTR-Directed Therapeutics*

The ICM method showed no positive dynamics against the background of tezacaftor/ivacaftor and ivacaftor (Figures 1 and 2). When the biopsies were stimulated with forskolin, no response was received either before taking the drug or against the background of long-term administration. The short-circuit current density (ΔISC) to the addition of carbachol and histamine changed to negative, which is characteristic of cystic fibrosis. Figures 1 and 2 show that ΔISC indicators are typical for patients with a severe genotype and lack of chloride channel function.

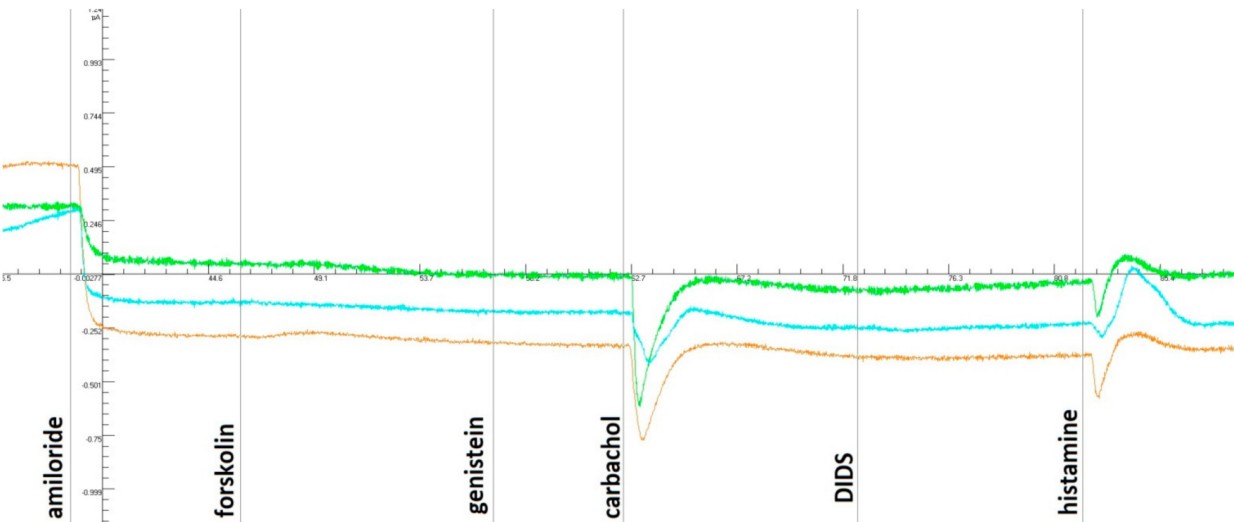

**Figure 1.** ICM measurements of the patient's rectal biopsies before taking a tezacaftor/ivacaftor and ivacaftor. With the introduction of amiloride, there was a decrease in the ΔISC and no response to forskolin/IBMX, and a change in the ΔISC in the negative direction was observed with the addition of carbachol and histamine.

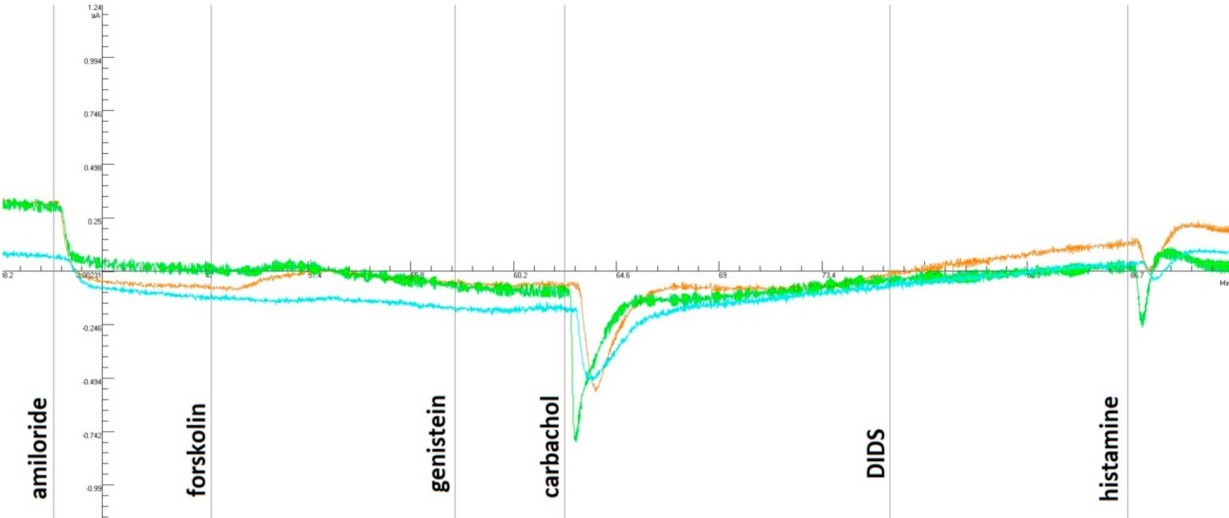

**Figure 2.** ICM measurements of the patient's rectal biopsies while *taking* tezacaftor/ivacaftor and ivacaftor. With the introduction of amiloride, there was a decrease in the ΔISC and no response to forskolin/IBMX, and a change in the ΔISC in the negative direction was observed with the addition of carbachol and histamine.

Due to the lack of dynamics of the condition against the background of targeted therapy and the absence of changes in the function of the chloride channel, according to the ICM data, it was assumed that a child with the F508del/F508del genotype had a complex allele.

To confirm the hypothesis of complex alleles, sequencing of all exons of the *CFTR* gene was performed. Along with the previously identified variant of p.Phe508del in the homozygous state (the frequency in the control sample of The Genome Aggregation Database (gnomAD) is 0.7172%), a second variant of the nucleotide sequence described earlier as pathogenic (CM063898) [16] was identified in exon 11 of the *CFTR* gene (chr7:117199524C>T), leading to the amino acid substitution of L467F (p.Leu467Phe, NM_000492.3) in a heterozygous state (Figure 3). The frequency of the identified variant of

the nucleotide sequence in The Genome Aggregation Database (gnomAD) control sample is 0.003892%. According to the literature [17], this variant may be part of a complex allele (in the cis position) with the pathogenic variant F508del (p.[Leu467Phe;Phe508del]). Both variants found were validated by Sanger sequencing.

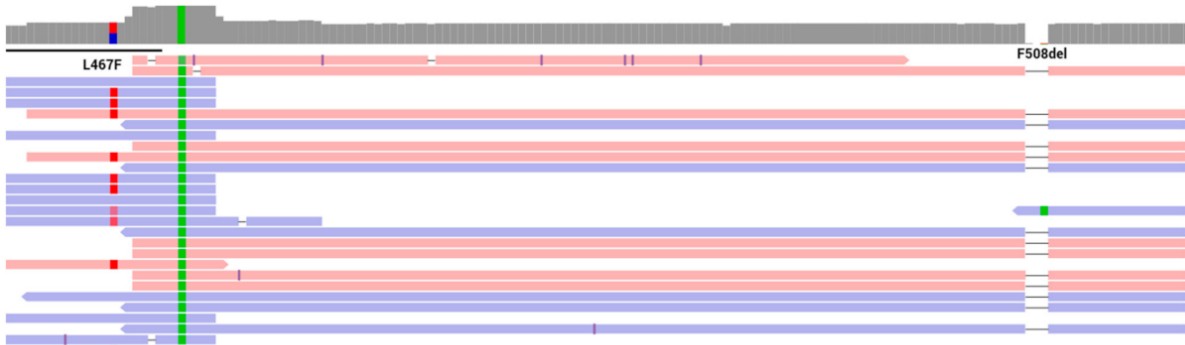

**Figure 3.** Variants L467F and F508del identified in the patient by mass-parallel sequencing in a heterozygous and homozygous state, respectively.

*3.3. Assessment of the Residual Functional Activity of CFTR Channel and the Effects of CFTR Modulators in Rectal Organoids Models*

A stable culture of intestinal organoids was obtained from rectal biopsies of the patient to assess the effect of ivacaftor, the combined effect of tezacaftor/ivacaftor and elexacaftor/tezacaftor/ivacaftor on the rescue of the functional CFTR protein, as well as to assess the residual function of the CFTR channel. Morphological features of the obtaining organoid culture indicate the loss of the functional CFTR protein. Organoids with the complex allele F508del/[L467F;F508del] (Figure 4) did not differ from the F508del/F508del culture and were characterized by a non-spherical shape, thickened walls and the absence of pronounced lumen, unlike the wt/wt control with a functional CFTR channel (Figure 4).

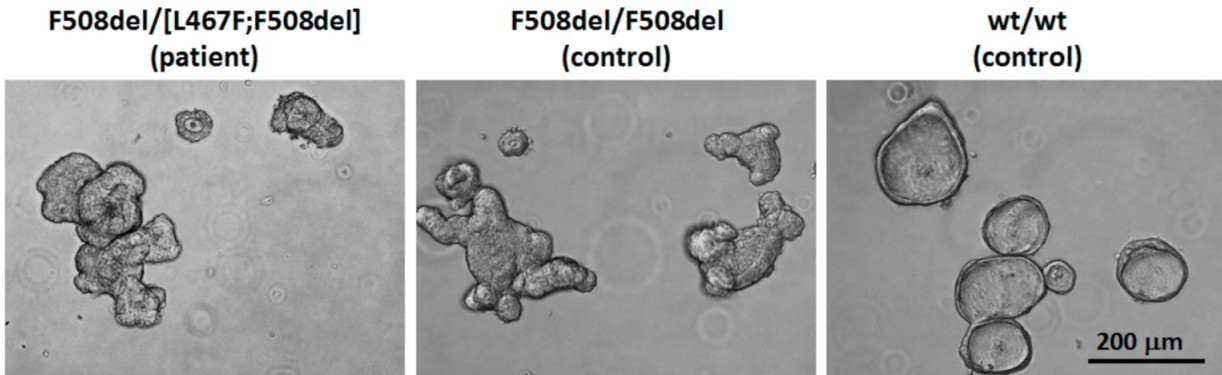

**Figure 4.** Morphological features of the culture of intestinal organoids with a complex allele [L467F;F508del]. Control—wt/wt and F508del/F508del organoids.

Stimulation by forskolin at a high concentration (5 µM, 1 h) of F508del/[L467F;F508del] organoid culture did not reveal the preservation of the residual CFTR function; organoids did not respond to swelling when exposed by forskolin (Figure 5). Compared with the F508del/F508del control culture, the AUC values were even lower: −20 ± 52 and 382 ± 106, respectively.

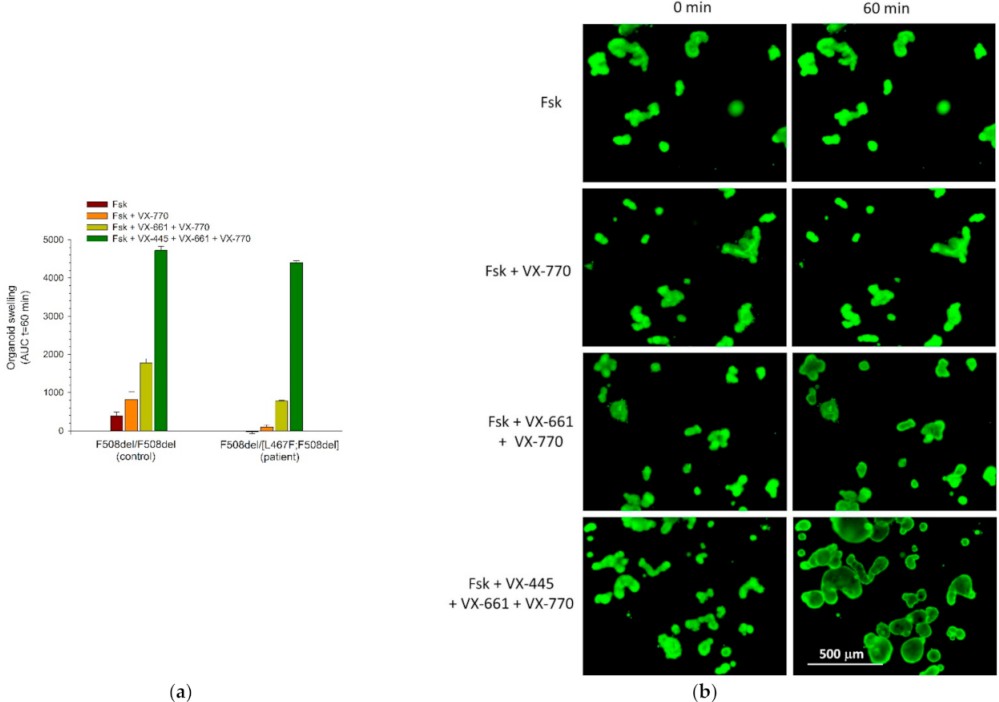

**Figure 5.** Assessment of the residual CFTR function and the effectiveness of the CFTR modulators (**a**) Quantitative evaluation of FIS results of intestinal organoids with a complex allele [L467F;F508del] in comparison with control F508del/F508del organoids. Fsk—forskolin, VX-770—ivacaftor, VX-661—tezacaftor, and VX-445—elexacaftor. (**b**) Representative images of intestinal organoids obtained from a patient with the genotype F508del/[L467F;F508del] before exposure to forskolin (5 μM) and after treatment. Concentration of all CFTR modulators—3.5 μM; staining—calcein (0.84 μM, 40 min), and scale bar—500 microns.

With the combined action of ivacaftor (VX-770) and forskolin, the swelling of intestinal organoids with a complex allele is insignificant and less than in the control F508del/F508del culture by a factor of 8.5. When combined with tezacaftor/ivacaftor (VX-770 + VX-661), there is a slight swelling of intestinal organoids in response to stimulation with forskolin; the AUC value is $775 \pm 21$. In the control, the response to tezacaftor/ivacaftor is $1773 \pm 113$ (Figure 5a).

Elexacaftor/tezacaftor/ivacaftor (VX-445 + VX-661 + VX-770) in a patient with the genotype F508del/(L467F;F508del) effectively restores the amount of functional protein CFTR on the apical membrane of epithelial cells of intestinal organoids (Figure 5a,b) since, when stimulated by forskolin, a high degree of swelling is observed; compared with their normal state, they increase by a factor of 2.3. The AUC values for the culture of organoids with a complex allele practically did not differ from the control values and were $4404 \pm 45$ and $4718 \pm 112$, respectively (Figure 5a).

### 3.4. Change in Clinical Status during Therapy with Elexafactor/Tezacaftor/Ivacaftor

A decision was made to switch the patient to the targeted drug Elexafactor/tezacaftor/ivacaftor (Trikafta®). Eleksafactor/tezacaftor/ivacaftor was started on 30 November 2022. A positive result of therapy was confirmed by a decrease in sweat test values (up to 55 mmol/L NaCl by March 2022, a decrease of 65 mmol/L), weight gain and improvement in respiratory function (Table 1).

Against the background of targeted therapy, in January 2022, the girl fell ill with COVID-19: fever up to 39.3 °C for 3 days, headaches, weight loss (PCR test dated 17 January 2022—positive); against this background, her dose of dornase alfa was doubled

and she took antipyretics as needed. Within a week, the symptoms were completely stopped. PCR test dated 27 January 2022 was negative.

In February 2022, she suffered from pneumonia (CT data, increased C-reactive protein); hemoptysis was noted.

In sputum, there was a massive growth of *Ps.aeruginosae* (mucoid phenotype), resistant to fluoroquinolones, amikacin, and a single growth of methicillin-sensitive *St.aureus*.

CT scan of the chest showed the course of right-sided poly-segmental pneumonia, possibly of viral origin. The percentage of lung involvement was 16%, with bronchiectases in the upper lobes of both lungs.

In addition to basic therapy, she received antibiotic therapy, tranexam, and targeted therapy with Elexacaftor/tezacaftor/ivacaftor was continued.

In April and May 2022, there were repeated cases of hemoptysis and acute viral infection. Antibacterial therapy was carried out according to plan, taking into account chronic *Pseudomonas aeruginosa* infection (Table 1).

In June 2022 (after 7 months of taking elexacaftor/tezacaftor/ivacaftor), condition was stable, weight—55.65 kg, height—176 cm, BMI—17.3 kg/m$^2$ (weight deficit with high growth). There were signs of the formation of "watch glasses" on the terminal phalanges of the fingers. On the skin of the forehead was back papulo-pustular acne. SatO$_2$—98%, RR—20 per min. On examination: no inflammatory changes in the blood, total immunoglobulin E-11 IU/mL (normal up to <100), vitamin D—18.0 ng/mL, sweat conductivity—66 mmol/L (decrease by 55 mmol/L), FEV1—104.3% (increase by 26.8% from the initial 82%).

## 4. Discussion

This clinical case demonstrates the lack of effect of CFTR therapy with tezacaftor/ivacaftor and ivacaftor as a modulator in a patient with the F508del/F508del genotype. The absence of a clinical effect was confirmed by the continuation of exacerbations with the need for antibacterial therapy and the lack of dynamics on the part of respiratory function. There was no increase in the functional activity of the CFTR (chloride) channel using a sweat sample and the ICM method. The assumption that the patient has an additional mutation in the cis position was confirmed by a complete analysis of the coding sequence of the *CFTR* gene. A complex allele [L467F;F508del] was identified on one of the patient's chromosomes, which, as shown, prevented the response to targeted therapy.

The frequency of the complex allele [L467F;F508del] is being studied and is 5/4617 [18] and 4/1524 [19]. According to the Research Centre for Medical Genetics in the Russian Federation, the frequency of the complex allele [L467F;F508del] is much higher and amounts to 8.2% among homozygotes according to the F508del variant (the results were not published at the time of preparation of the article). The studies of Baatallah et al. have shown that patients carrying complex alleles [L467F;F508del] in combination with F508del respond worse to treatment with correctors and/or potentiators when using a combination of VX-809 + VX-770 (lumacaftor/ivacaftor) than patients with the genotype F508del/F508del [20]. The importance of determining the complete genotype of patients with cystic fibrosis and, especially, identifying complex alleles in patients carrying the F508del mutation to predict the response to treatment and confirm the effectiveness of new combined complex CFTR modulators is emphasized [20]. Further functional evaluation of complex alleles is necessary, in terms of their effect on the activity of the CFTR channel, to clarify their pathogenicity. The literature describes complex alleles of the CFTR gene that do not affect the severity of the disease and the response to targeted therapy, but there are also complex alleles with unclear clinical significance [4].

The results obtained by the ICM method were confirmed on the model of intestinal organoids. When stimulated with forskolin, the organoids of a patient with the specified genotype F508del/[L467F;F508del] did not respond with swelling, which indicates a complete loss of CFTR function. The treatment of intestinal organoids with tezacaftor/ivacaftor (VX-661 + VX-770) causes a slight increase in the amount of functional protein CFTR (AUC values < 1000), while the organoids of the control culture F508del/F508del respond

with strong swelling (AUC values~2000). When the AUC value is lower than 1000, therapy with CFTR modulators cannot be recommended to patients since their appointment will not lead to a visible therapeutic effect [21]. Thus, the data obtained on the effect of tezacaftor/ivacaftor on the rescue of CFTR function in a patient with a complex allele [L467F;F508del] are consistent with the clinical results of tezacaftor/ivacaftor therapy, which did not have a positive effect. At the same time, this method demonstrated the sensitivity of the chloride channel to the drug elexacaftor/tezacaftor/ivacaftor.

In the 20 years since the first description of complex alleles, a significant amount of new knowledge about them has accumulated. However, further research in this direction is needed. This case shows a personalized approach to therapy with CFTR modulators of a patient with CF, which includes the most complete molecular genetic study using sequencing to search for complex alleles, the ICM method, and the FIS assay on intestinal organoids. A personalized approach is especially important for patients who do not have a response to therapy with CFTR modulators. The drug of the second generation of CFTR modulators—elexacaftor/tezacaftor/ivacaftor (VX-445 + VX-661 + VX-770)—prescribed after an additional examination on intestinal organoids, showed a clinical effect of 16%, despite COVID-19, lung damage, and the presence of chronic aspergillosis. The clinical efficacy was confirmed by the results of a sweat test and spirometry.

First-generation drugs can help people with two F508del mutations, which is only about 50 percent of people with cystic fibrosis worldwide and 30% in the Russian Federation, according to the 2020 registry [22]. Since almost 90% of cystic fibrosis patients have one or two F508del mutations, next-generation modulators can be used in a much larger number of people with cystic fibrosis. The first modulator of the second generation is a combination of three modulators—trikafta (elexacaftor/tezacaftor/ivacaftor)—for the treatment of patients aged 6 years and older, with mutations occurring in 90% of cases. The appearance of a new corrector, elexacaftor (VX-445), significantly increased the effectiveness of therapy and improved the safety profile [23]. Results of the Phase 3 study [24] showed that the use of elexacaftor/tezacaftor/ivacaftor in people with the F508del/F508del genotype who had previously received tezacaftor and ivacaftor also demonstrated an increase in $FEV_1$ by 10.0% from the baseline level at week 4, compared with the control group that received a placebo ($p < 0.0001$). At the same time, the safety profile for the triple combination with VX-445 was characterized by significantly fewer undesirable side reactions, especially in the respiratory tract. In the above clinical case, the patient's $FEV_1$ increased from 82% to 106% (by 29.3% of the initial one), the sweat sample decreased by 55 mmol/L and the composition was 55–65 mmol/L. Monitoring biochemical parameters and the state of the organ of vision revealed no pathology.

Previously, Sondo et al. used ex vivo nasal epithelium models to study the effect of elexacaftor/tezacaftor/ivacaftor on CFTR function in two patients carrying the complex allele [L467F;F508del] in a compound with class I mutations G542X and E585X [25]. The results showed no response when exposed to a triple combination of CFTR modulators. Using heterologous protein expression and Western blotting, the authors found that the complex allele [L467F;F508de] leads to a decrease in the formation of the total amount of CFTR protein, while only an immature form of the CFTR protein is formed, devoid of functional activity. The use of CFTR modulators elexacaftor/tezacaftor/ivacaftor does not increase the amount of mature fully glycosylated form of CFTR; therefore, it does not affect the activity of the [L467F;F508del]-CFTR protein [25].

## 5. Conclusions

The described clinical case demonstrates the absence of the effect of CFTR therapy with tezacaftor/ivacaftor and ivacaftor in a patient with the F508del/F508del genotype. The assumption that the patient has an additional mutation in the cis position was confirmed during sequencing of the *CFTR* gene and a complex allele was identified [L467F;F508del]. This case indicates the need to search for additional mutations in cases of the ineffectiveness of targeted therapy by sequencing the entire *CFTR* gene. The results were obtained by the

ICM method and on the model of intestinal organoids, confirming that the CFTR channel in a patient with the genotype F508del/[L467F;F508del] is not functional. A weak effect of tezacaftor/ivacaftor on the restoration of CFTR function was found on the model of intestinal organoids of the patient, in contrast to patients homozygous for the genetic variant F508del and without other mutations in the cis position. Based on the positive results of the FIS assay, the patient was recommended therapy with a combination drug—elexacaftor/tezacaftor/ivacaftor (VX-445 + VX-661 + VX-770). The use of the drug for 7 months led to a significant improvement in the clinical picture of the patient.

**Author Contributions:** E.K., S.K. and D.G. devised the aim, planned and designed the research. A.E., Y.M. and T.B. planned the experiments. V.S., A.V., I.A. and E.K. evaluated the patient. N.B. and Y.M. performed the experiments with ICM and intestinal organoids. A.P., T.A. and V.K. performed genetic analysis. E.K., A.E., Y.M. and V.S. wrote the original draft. A.P., D.G., E.Z. and V.S. reviewed and edited the original draft. All authors have read and agreed to the published version of the manuscript.

**Funding:** This research was funded by the Russian Science Foundation (grant No.22-15-00473, "Investigation of the effect of complex alleles of the CFTR gene on the functional activity of the chloride channel for personalized selection of targeted therapy for cystic fibrosis").

**Institutional Review Board Statement:** The study and informed voluntary consent form were approved by the Ethics Committee at "Research Centre for Medical Genetics" on 15 October 2018 (Chairperson of the Ethics Committee–Prof. L.F. Kurilo).

**Informed Consent Statement:** Informed consent was obtained from all patients or from their parents enrolled in the study.

**Data Availability Statement:** The datasets used and/or analysed during the current study are available from the corresponding author upon reasonable request.

**Conflicts of Interest:** The authors declare no conflict of interest.

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
