# Peer review of "Personalized Selection of a CFTR Modulator for a Patient with a Complex Allele [L467F;F508del]"

_cimb, doi:10.3390/cimb44100349_

Round 1

Reviewer 1 Report

Thank you for submitting this interesting case to the Journal. This is a highly  valuable observation with a comprehensive evaluation. I suggest only minor revisions ;

1. Results: First three paragraphs should be re-written in ordinary sentence forms.

2. Table 1 should be simplified, no need to give all details, grouping them will be better and easy to follow.

3. Page 12, line 368, "Macrodact"  should be "Macroduct".  

Author Response

  1. Results: First three paragraphs should be re-written in ordinary sentence forms.– made corrections
  2. Table 1 should be simplified, no need to give all details, grouping them will be better and easy to follow. - shortened and simplified
  3. Page 12, line 368, "Macrodact" should be "Macroduct".  – corrected.

Reviewer 2 Report

The article “Personalized selection of a CFTR modulator for a patient with a complex allele L467F/F508del” by Elena Kondratyeva et al presents an interesting Cystic Fibrosis case. The methodology used by the authors is accurate. The results presented in this article show the importance of personalized functional CFTR studies in vitro, and how the results of these studies correlate with the clinics. Thus, functional assays such as the ones presented here, can predict the most appropriate therapeutic option for each patient with CF. However, the article requires changes in the structure and an extensive editing style in Introduction, Methods and Results. 

Introduction:

More information and background are needed in the Introduction section: 

-I suggest adding a brief background about how CFTR variants are classified into different classes.

-It is necessary to add background about modulator treatments in CF, specially luma/iva therapy in CF disease.

Paragraph 48-55: consider adding more background about the methodologies mentioned, the ICM measurements and FIS assay

Materials and Methods:

-There should be a specific section about how the rectal/intestinal samples were obtained. 

- The explanation of the “obtaining intestinal organoids” can be shorter if no changes were made from the original protocol referenced.

-Line 178: Matrigel dilution

-Line 197: authors should briefly mention how they passaged the organoids (references can be added).

-Line 199 and 201: concentration used for modulator drugs should be referenced or explained. 

-FIS assay and posterior analysis can be referenced.

-How many healthy volunteers and F508del/F508del samples were used for this study?

-Line 291 and 292: figures from these results should be indicated.

-Line 299-300: figure(s) from this/these result(s) should be indicated

-Figure 5 and 6: I suggest to combined both figures

-Lines 314-319: reference a figure. 

-Lines 320-326: reference a figure.

Results 

-Results presents an undefined structure. 

-The clinical history of the patient (lines 212-246, lines 283-286, Table 1) should be presented, in a shorter/resumed version, in the introduction section instead than in the results section. 

-The writing style of the Results sections should be reviewed – needs to be more accurate and formal. 

-Adding subheadings in the Results sections would help the reader to properly follow all the data presented here.

-Information in lines 327-371 is very dense – style must be reviewed, and information should be simplified or resume in a Table.

No comments for Discussion and Conclusions sections.

Author Response

  1. Introduction: - added to text

More information and background are needed in the Introduction section: 

-I suggest adding a brief background about how CFTR variants are classified into different classes.

-It is necessary to add background about modulator treatments in CF, specially luma/iva therapy in CF disease.

  1. Paragraph 48-55: consider adding more background about the methodologies mentioned, the ICM measurements and FIS assay - added information to text

  1. Materials and Methods:

-There should be a specific section about how the rectal/intestinal samples were obtained. added information to text

- The explanation of the “obtaining intestinal organoids” can be shorter if no changes were made from the original protocol referenced - reduced

-Line 178: Matrigel dilution - proposal is correct

-Line 197: authors should briefly mention how they passaged the organoids (references can be added). - added, corrected

-Line 199 and 201: concentration used for modulator drugs should be referenced or explained. - provided a link to the article by J. Beekman and A. Vonk

-FIS assay and posterior analysis can be referenced. - references added

-How many healthy volunteers and F508del/F508del samples were used for this study? provided information. As a control were used two organoid cultures of CF-patient with F508del/F508del genotype and two organoid cultures of non-CF individuals

-Line 291 and 292: figures from these results should be indicated. - have corrected

-Line 299-300: figure(s) from this/these result(s) should be indicated - have corrected

-Figure 5 and 6: I suggest to combined both figures - changed

-Lines 314-319: reference a figure. - added

-Lines 320-326: reference a figure. - added

  1. Results 

-Results presents an undefined structure. - corrected

-The clinical history of the patient (lines 212-246, lines 283-286, Table 1) should be presented, in a shorter/resumed version, in the introduction section instead than in the results section. - we thank the reviewer for the comment, but we believe that the description of the clinical picture is better left in the results, so that it is more convenient to compare the results before and after Trikaft therapy.

-The writing style of the Results sections should be reviewed – needs to be more accurate and formal. - corrected

-Adding subheadings in the Results sections would help the reader to properly follow all the data presented here. - added headings for all sections

-Information in lines 327-371 is very dense – style must be reviewed, and information should be simplified or resume in a Table. - shortened, corrected
